# Using Mindful Attention Awareness Scale on male prisoners: Confirmatory factor analysis and Rasch models

Ali Poorebrahim[1], Chung-Ying Lin[2]*, Vida Imani[3], Shapour Soltankhah Kolvani[4], Seyed Abbas Alaviyoun[4], Narges Ehsani[5], Amir H. Pakpour[5,6]*

1 Student Research Committee, Qazvin University of Medical Sciences, Qazvin, Iran, 2 Institute of Allied Health Sciences and Departments of Occupational Therapy and Public Health, National Cheng Kung University Hospital, College of Medicine, National Cheng Kung University, Tainan, Taiwan, 3 Pediatric Health Research Center, Tabriz University of Medical Sciences, Tabriz, Iran, 4 Guilan Prison Research Council, Rasht, Guilan, Iran, 5 Social Determinants of Health Research Center, Research Institute for Prevention of Non-Communicable Diseases, Qazvin University of Medical Sciences, Qazvin, Iran, 6 Department of Nursing, School of Health and Welfare, Jönköping University, Jönköping, Sweden

* pakpour_amir@yahoo.com, apakpour@qums.ac.ir (AHP); cylin36933@gmail.com (CYL)

**Data Availability Statement:** All relevant data are within the paper and its Supporting information files.

## Abstract

### Aim

This study tested the construct validity (i.e., factor structure) of the Persian Mindful Attention Awareness Scale (MAAS) on a sample of male prisoners.

### Methods

All the participants (mean±SD age = 39.44±7.94 years) completed three scales—the Persian MAAS, the Insomnia Severity Index (ISI), and the 12-item General Health Questionnaire (GHQ-12). Confirmatory factor analysis (CFA) and Rasch analysis with differential item functioning (DIF) were applied to examine the construct validity of the MAAS. Specifically, the DIF was tested across different insomnia status (using ISI with a cutoff of 15), psychiatric well-being status (using GHQ-12 with a cutoff of 12), and age (using mean age of 39.44 as the cutoff).

### Results

The CFA results showed a single factor solution for the Persian MAAS. The Rasch results showed all MAAS items fit in the construct (infit mean square [MnSq] = 0.72 to 1.41; outfit MnSq = 0.74 to 1.39) without displaying DIF items (DIF contrast = -0.34 to 0.31 for insomnia condition; -0.22 to 0.25 for psychiatric well-being; -0.26 to 0.29 for age).

### Conclusions

The Persian version of the MAAS is, therefore, a valid instrument to measure mindfulness among Iranian male prisoners.

**Funding:** The authors received no specific funding for this work.

**Competing interests:** The authors have declared that no competing interests exist.

**Abbreviations:** CFI, Confirmatory factor analysis; infit, Information-weighted fit statistic; MnSq, Mean square; outfit, Outlier-sensitive fit statistic; RMSEA, Root mean square error of approximation; SEM, Structural equation model; SRMR, Standardized root mean square residual; TLI, Tucker-Lewis index.

# Introduction

Studies investigating mindfulness have been increased and keep growing in the recent years [1–4]. Specifically, researchers tried to identify how attentional present-centeredness; i.e., a central facet of "mindfulness" [5], or in converse, how inattentiveness or "mind-wandering" [6, 7], may relate to psychological health among different populations. Based on the empirical findings from correlational, interventional, and laboratory research, a conclusion has been made that mindfulness can be beneficial toward psychological well-being, including increased well-being, reduced symptoms, and improved behavioral regulation [8]. Therefore, mindfulness-based interventions (MBIs) have been designed and developed with promising and positive results across different conditions, including sexual functions [9], generalized anxiety disorder [10], and diabetes [11]. Moreover, a review summarized the effectiveness of MBIs through evaluating randomized controlled trials (RCTs) and found that the analyzed rigorous RCTs showing the effectiveness of MBIs on different outcomes, including addition, chronic pain, and depression relapse [12].

In order to maximize the benefits of MBIs, using a validated instrument to understand mindfulness is essential. The concept of "mindfulness" has been argued for its operational definition [13] because of the arguments on whether mindfulness is a single factor [1, 14, 15] or a multifactor [5, 16] construct. However, current evidence on mindfulness models is prone to multifactor [17, 18]. Nevertheless, a single factor (i.e., attention and awareness of the present) can be used as the basis to assess mindfulness before expanding the concept of mindfulness to pose various dimensionalities, considering that a single-factor structure is easier to be studied and understood than a multi-factor structure. Therefore, the self-reported Mindful Attention Awareness Scale (MAAS), which has a one-factor structure, is one of the potential instruments for healthcare providers to assess mindfulness [1]. Although the MAAS suffers from some problems in its content validity (i.e., only assessing part of the mindfulness via mindlessness facet) and becomes outdated in the Western countries, it deserves a breaking point for the countries without appropriate instruments assessing mindfulness.

The MAAS was developed with a name of "trait scale" because the developers hypothesized that inattentiveness is a psychological trait that can be independently measured in the general population [1]. More specifically, the MAAS assesses the tendency to be attentive and aware of present moment experience in daily living for people who do not have any meditation experience [1]. Given that the promising psychometric properties of the MAAS have been reported across different populations (e.g., [1, 4, 19, 20]) and different languages (e.g., [21–23]), the MAAS has been widely used in both research and clinical settings [3]. Additionally, several brief versions of the MAAS have been developed and validated (e.g., [24, 25]). In addition, the MAAS has been integrated into other mindfulness instruments to assess the nature of multi-factor structure in mindfulness (e.g., the Five Facet Mindfulness Questionnaire) [26].

However, it is unclear whether the MAAS can be validly applied to prisoners because to the best of our knowledge, no psychometric information has been reported for this specific population. A sound instrument needs to have accumulated evidence on its psychometric testing across different populations to fulfill the scientific inquiry [27]. More specifically, the evidence of an instrument's psychometric properties is highly dependent on the tested populations; therefore, the satisfactory reliability and validity found in one population (e.g., adolescents) cannot be simply generalized to another population (e.g., elders). Hence, it is crucial for the MAAS to be tested for its psychometric properties in a population that has never been examined (i.e., the prisoners used in the present study).

Prisoners are proposed to have high risk of developing mental illness or to have high chance to worsen their mental health problems prior to imprison during imprisonment [27].

Specifically, the development of or worsened psychological disorder may be contributed by several factors during imprisonment, such as overcrowding or lack of privacy, meaningless activities or imposed loneliness, behavioral issues of employees and fellow prisons, or insecurity about future [28]. In this regard, the imprisonment factors may impact the prisoners to rate their mindfulness or to interpret the content of the MAAS items. Therefore, assessing the construct validity of the MAAS among this specific population is warranted.

The aim of this study was to use two theories of psychometric testing to examine the construct validity of the Persian MAAS among male prisoners. Confirmatory factor analysis from the classical test theory was used to examine whether the Persian MAAS has a one-factor structure in the male prisoners. Then, Rasch analysis from the item response theory was applied to evaluate (1) whether all the Persian MAAS items are embedded in the same construct (i.e., mindfulness); (2) whether the Likert-type scale used in the Persian MAAS follow its difficulty order (i.e., score 1 should be easier than score 2); and (3) whether the Persian MAAS was interpreted similarly across different conditions (insomnia, psychiatric well-being, and age) in the male prisoners.

## Methods

### Participants and procedure

The cross-sectional study was conducted in the main prison in Guilan province (Lakan Prison) between February 2019 and December 2019. The prison was created in 1990 with 23,000 square meters of infrastructure and about 7 hectares of enclosed area. It is located 13 km from Rasht city center. During the data collection period, the total number of prisoners was 3200 with majority of the inmates were males. The target participants of the present study were drug-abusing prisoners in Lakan prison and 600 of 1400 male prisoners were randomly selected for assessing their eligibility. Eligible participants should achieve the inclusion criteria of (1) male offenders currently serving a prison sentence with problem drug abuse, (2) aged 18 years of order, and (3) were able and willing to provide informed consent. Prisoners were excluded if having cognitive impairment from conditions such as severe illness or injury.

Ethics approvals for this study were granted by the Regional Research Ethics Committee at Qazvin University of Medical Sciences (IR.QUMS.REC.1397.294) and Prisons and Security and Corrective Measures Organization (70/31/13). The participation to the study was voluntary and anonymous. The study was performed in accordance with the Declaration of Helsinki. Written informed consent was obtained from participants before enrollment.

### Instruments

**Mindful Attention Awareness Scale (MAAS).** The MAAS contains 15 items to assess individual differences in the level of mindfulness and it is one of the commonly used instruments in assessing mindfulness [1, 3]. All the items are rated on a 6-point Likert-type scale and a summated score can be computed for the MAAS total score, where a higher score indicates a higher level of mindfulness. The MAAS has been translated into different languages with promising psychometric properties [19, 22, 23, 29, 30], including Persian versions [31, 32]. Moreover, the internal consistency of the MAAS was adequate (Cronbach's $\alpha = 0.878$; McDonalds' $\omega = 0.879$).

**Insomnia Severity Index (ISI).** The ISI contains 7 items to assess severity of the insomnia. All the items are rated on a 5-point Likert-type scale and a summated score can be computed for the ISI total score (ranged between 0 and 28), where a higher score indicates a higher level of insomnia. Moreover, the total score of ISI ranges between 0 and 28, where 0–7 indicates absence of insomnia, 8–14 indicates sub-threshold insomnia, 15–21 indicates moderate

insomnia, and 22–28 indicates severe insomnia [33]. The ISI has been translated into Persian version with promising psychometric properties (e.g., Cronbach's α = 0.82 to 0.87) [34, 35]. Moreover, the internal consistency of the ISI was adequate (Cronbach's α = 0.900; McDonalds' ω = 0.902).

**12-item General Health Questionnaire (GHQ-12).**   The GHQ-12 contains 12 items to assess health, especially psychiatric well-being, of an individual. All the items are rated on a 4-point Likert-type scale and a summated score can be computed for the GHQ-12 total score, where a higher score indicates worse psychiatric well-being. Moreover, the total score of GHQ-12 ranges between 0 and 36, where a score equal to and greater than 12 indicates having psychiatric well-being problem [36]. The GHQ-12 has been translated into Persian version with promising psychometric properties (e.g., Cronbach's α = 0.78 to 0.84) [37, 38]. Moreover, the internal consistency of the GHQ-12 was adequate (Cronbach's α = 0.714; McDonalds' ω = 0.653).

## Data analysis

Apart from the descriptive statistics (mean and SD for continuous variables; frequency and percentage for categorical variables), confirmatory factor analysis (CFA) using lavaan package in the R software and Rasch analysis using WINSTEPS 4.3.0 (winsteps.com) were used to test the construct validity of the Persian MAAS among male prisoners.

In the CFA, a one-factor structure was tested for the MAAS using the diagonally weighted least squares (DWLS) estimator. Several fit indices, including comparative fit index (CFI), Tucker-Lewis index (TLI), root mean square error of approximation (RMSEA), and standardized root mean square residual (SRMR), were applied to examine whether the one-factor structure is supported. Specifically, both CFI and TLI > 0.9 together with both RMSEA and SRMR < 0.08 support the one-factor structure of the MAAS [39]. Factor loadings of the MAAS items were evaluated using the cutoff of 0.4; that is, a factor loading > 0.4 indicates the need of that item [40].

In the Rasch, a rating scale model (RSM) was used to examine the MAAS. We considered using Rasch model with RSM instead of other item-response theory models (e.g., the two parameter or the three logistic parameter model with partial credit model or graded response model) because Rasch model with RSM can provide simpler estimation in modeling (i.e., Rasch model with RSM needs not to estimate other parameter like discrimination and the category difference for every two responses). Thus, the benefit of using such a model than other types of item-response theory model is it fits better with the principle of parsimony.

In the Rasch modeling with RSM, global test on type I error rates across all item fits was performed first, and a nonsignificant test indicates all items embedded in the same construct. Then, mean square (MnSq) of information-weighted fit statistics (infit) and that of outlier-sensitive fit statistics (outfit) were applied to understand whether an item fit in the MAAS construct. Specifically, both infit and outfit MnSq range between 0.5 and 1.5 indicate the fit of an item [39]. Moreover, the 6-point Likert-type of the MAAS was examined whether the score increased monotonically; that is, the difficulty of a lower score is not greater than a higher score. Average measure of the six scores and step measure of every two scores were used. Additionally, both infit and outfit MnSq were used to examined whether the monotonic increase was supported. Specifically, both infit and outfit MnSq range between 0.4 and 1.6 indicate the monotonic increase [41]. Local independence was tested to understand whether residual correlations exist among items. Specifically, we tested the Rasch residual for every item and used the correlations between the Rasch residuals to examine the local independence, where an absolute correlation coefficient > 0.4 indicating substantial dependence. Lastly, differential

item functioning (DIF) across for the thresholds were used to examine whether the male prisoners under different conditions (i.e., different types of health problems and different age) interpret the MAAS items similarly. Specifically, the DIF was examined across insomnia condition (using the cutoff of 15 in the ISI), psychiatric well-being (using the cutoff of 12 in the GHQ-12), and age group (using the mean age of 39.44). A DIF contrast < 0.05 indicates that different groups interpret a MAAS item similarly [42, 43].

Apart from the CFA and Rasch, Spearman's rho correlations were carried to understand the associations between the MAAS, the ISI, and the GHQ-12 scores.

## Results

Table 1 demonstrates the characteristics of the sample. In brief, the mean (SD) age of the male prisoners was 39.44 (7.94) years. They received, on average, 9.51 (3.05) years of education. More than half of the participants were married (58.4%). The majority of the participants had a history of drug abuse (78.9%) and more than half of the participants had a history of alcohol abuse (59.3%). On average, the MAAS mean score was 4.21 (0.91).

The one-factor structure of the MAAS is supported by the fit indices of the CFA, including CFI (0.982), TLI (0.979), RMSEA (0.044), 95% CI of RMSEA (0.032, 0.055), and SRMR (0.064), except for the significant $\chi^2$ test (Table 2). Moreover, all the factor loadings in the MAAS were strong (0.478 to 0.679) and significant (Table 3). Rasch analysis further shows that the global test on type I error rates associated with the Rasch item fit statistics was nonsignificant (p = 0.48), indicating the satisfactory item fit across all the items. Indeed, all the items fit in the same construct (infit MnSq = 0.72 to 1.41; outfit MnSq = 0.74 to 1.39) with all items

**Table 1. Socio-demographic characteristics of prisoners in Rasht Lakan correctional institution.**

|  | Mean or n | SD or % |
|---|---|---|
| Age (year) | 39.44 | 7.94 |
| Educational year | 9.51 | 3.01 |
| Marital status |  |  |
| Single | 88 | 24.4% |
| Married | 211 | 58.4% |
| Divorced | 62 | 17.2% |
| Having children |  |  |
| Yes | 199 | 55.1% |
| No | 162 | 44.9% |
| Occupational status |  |  |
| Jobless | 10 | 2.8% |
| Daily labor | 137 | 37.9% |
| Self-employed | 200 | 55.4% |
| Employed | 14 | 3.9% |
| History of drug abuse |  |  |
| Yes | 285 | 78.9% |
| No | 76 | 21.1% |
| History of alcohol abuse |  |  |
| Yes | 214 | 59.3% |
| No | 147 | 40.7% |
| MAAS mean score | 4.21 | 0.91 |

MAAS = Mindful Attention Awareness Scale

**Table 2. Confirmatory factor analysis (CFA) results.**

| Fit indices | One factor |
|---|---|
| $\chi^2$ (*df*)/ p-value | 151.313 (90)/ < 0.001 |
| Normed $\chi^2$ | 1.68 |
| CFI | 0.982 |
| TLI | 0.979 |
| RMSEA (90%CI) | 0.044 (0.031, 0.055) |
| SRMR | 0.064 |

Normed $\chi^2$ = $\chi^2$ divided by *df*.

CFI = comparative fit index; TLI = Tucker-Lewis index; RMSEA = root mean square of error approximation; SRMR = standardized root mean square residual; CI = confidence interval.

were mutually correlated (bi-point serial correlations = 0.51 to 0.60) (Table 3). Additionally, the ordering of the six-point Likert scale was monotonically increased in the difficulty (average measure from -0.46 at threshold 1 to 1.09 at threshold 6; step measure was -0.97 at threshold 2 and 0.80 at threshold 6; infit MnSq = 0.89 to 1.06 and outfit MnSq = 0.81 to 1.14 for the thresholds) (Table 4). Tests on local response dependence indicate that there were no substantial residual correlations among the items (r = -0.33 to 0.27).

No DIF items were displayed for the MAAS across insomnia condition (DIF contrast = -0.34 to 0.31), across psychiatric well-being (DIF contrast = -0.22 to 0.25), and across age (DIF = -0.26 to 0.29) (Table 5). Moreover, the MAAS total score was significantly correlated with the ISI (r = -0.601; p<0.001) and the GHQ-12 (r = -0.384; p<0.001) scores.

## Discussion

The present study, to the best of our knowledge, is the first one that assessed the psychometric properties of the MAAS on a specific population: prisoners. Our findings supported the construct validity of the Persian MAAS; that is, the Persian MAAS is interpreted as a one-factor structure among male prisoners. Furthermore, prisoners under different conditions (i.e., different age, different psychiatric well-being, and different sleep problems) all interpret the Persian MAAS similarly. Therefore, the use of MAAS for prisoners is verified to be valid, including combing and comparing the level of mindfulness across different conditions for prisoners. Moreover, the MAAS mean score found in the present sample of prisoners was slightly higher than that of college students (MAAS mean score = 3.89 for Bangla students [19]; 3.88 for Greek students [44]; and 3.72 to 4.01 for American students [1, 30, 45]) but lower than that of general population (MAAS mean score = 4.86 for Italians [23] and 4.45 for Canadians [46]).

As comparing with prior studies on MAAS psychometric properties [19, 22, 23, 29, 30], our results share similar CFA results, which indicate that the MAAS is valid in a one-factor structure. However, a previous Iran study showed that the Persian MAAS has a two-factor structure [32]. A possible reason for the different structure findings may be due to the studied samples. Mohsenabadi et al. [32] examined the psychometric properties of the MAAS on a sample of adolescents aged between 12 and 18 years. On the other hand, prior studies showing the one-factor structure of MAAS examined its psychometric properties on young adults (e.g., university students) [19, 22, 23, 29, 30]. Because the cognition developments are different between adolescents and young adults, it is possible that different structures of MAAS are interpreted between adolescents and young adults. Given that the mean age of our participants was 39.44;

**Table 3. Standardized factor loadings in confirmatory factor analysis (CFA) and Rasch difficulties and fit statistics for each item.**

| MAAS Items description | CFA Loadings[a] | Difficulties | Item discrimination | Rasch Infit MnSq | Outfit MnSq | Model SE | Correlation |
|---|---|---|---|---|---|---|---|
| 1. I could be experiencing some emotion and not be conscious of it until sometime later | 0.517 | -0.36 | 1.09 | 0.72 | 0.80 | 0.05 | 0.51 |
| 2. I break or spill things because of carelessness, not paying attention, or thinking of something else. | 0.600 | -0.66 | 1.04 | 1.11 | 0.99 | 0.06 | 0.54 |
| 3. I find it difficult to stay focused on what's happening in the present. | 0.598 | 0.20 | 1.07 | 0.97 | 1.08 | 0.04 | 0.56 |
| 4. I tend to walk quickly to where I'm going without paying attention along the way. | 0.491 | 0.17 | 0.77 | 1.14 | 1.18 | 0.04 | 0.52 |
| 5. I tend not to notice feelings of physical tension or discomfort until they really grab my attention | 0.478 | 0.36 | 0.69 | 1.19 | 1.21 | 0.04 | 0.52 |
| 6. I forget a person's name almost as soon as I've been told it for the first time. | 0.504 | 0.21 | 0.71 | 1.41 | 1.39 | 0.04 | 0.51 |
| 7. It seems I'm "running on automatic" without much awareness of what I'm doing. | 0.563 | 0.05 | 1.05 | 1.02 | 1.01 | 0.04 | 0.57 |
| 8. I rush through activities without being really attentive to them. | 0.627 | 0.01 | 1.18 | 0.81 | 0.74 | 0.04 | 0.60 |
| 9. I get so focused on the goal I want to achieve that I lost touch with what I am doing right now to get there | 0.546 | 0.11 | 0.97 | 0.93 | 0.97 | 0.04 | 0.55 |
| 10. I do jobs or tasks automatically, without being aware of what I'm doing | 0.598 | 0.09 | 0.97 | 1.05 | 1.08 | 0.04 | 0.58 |
| 11. I find myself listening to someone with one ear, doing something else at the same time. | 0.545 | 0.12 | 0.85 | 0.93 | 1.00 | 0.04 | 0.53 |
| 12. I drive places on "automatic pilot" and then wonder why I went there. | 0.679 | -0.38 | 1.12 | 1.00 | 0.90 | 0.05 | 0.59 |
| 13. I find myself preoccupied with the future or the past | 0.643 | 0.42 | 1.29 | 0.86 | 0.83 | 0.04 | 0.62 |
| 14. I find myself doing things without paying attention. | 0.617 | -0.06 | 1.15 | 0.94 | 0.87 | 0.04 | 0.58 |
| 15. I snack without being aware that I'm eating. | 0.511 | -0.27 | 0.89 | 1.15 | 1.09 | 0.05 | 0.52 |

[a] Loadings are derived from single-factor model.

Infit = information-weighted fit statistic; Outfit = outlier-sensitive fit statistics; MnSq = mean square.

MAAS = Mindful Attention Awareness Scale; CFA = confirmatory factor analysis

our participants might have similar interpretation of the MAAS structure to the structure interpretation from young adults instead of that from adolescents.

Some studies used Rasch analysis to examine the MAAS found that the MAAS has some misfit items [20, 47]. In contrast, our findings indicated that all the MAAS items were fit in the

**Table 4. Response disordering tests on Mindful Attention Awareness Scale.**

| | Average measure | Step measure | Infit MnSq | Outfit MnSq |
|---|---|---|---|---|
| Score 1 | -0.46 | - - | 1.06 | 1.14 |
| Score 2 | -0.15 | -0.97 | 1.01 | 1.03 |
| Score 3 | 0.05 | -0.60 | 0.99 | 0.99 |
| Score 4 | 0.30 | 0.30 | 0.89 | 0.81 |
| Score 5 | 0.62 | 0.48 | 0.94 | 0.92 |
| Score 6 | 1.09 | 0.80 | 1.09 | 1.06 |

1: *Almost always*; 2: *Very frequently*; 3: *Somewhat frequently*; 4: *Somewhat infrequently*; 5: *Very infrequently*; 6: *Almost never*.

Infit = information-weighted fit statistic; Outfit = outlier-sensitive fit statistics; MnSq = mean square.

**Table 5. Differential item functioning (DIF) across different subgroups in the Mindful Attention Awareness Scale.**

| Item # | DIF across insomnia condition[a] | DIF across psychiatric disorder[b] | DIF across age[c] |
|---|---|---|---|
| 1. I could be experiencing some emotion and not be conscious of it until sometime later | 0.08 | 0.13 | 0.01 |
| 2. I break or spill things because of carelessness, not paying attention, or thinking of something else. | -0.22 | 0.25 | 0.01 |
| 3. I find it difficult to stay focused on what's happening in the present. | -0.15 | 0.01 | 0.15 |
| 4. I tend to walk quickly to where I'm going without paying attention along the way. | 0.16 | 0.09 | -0.12 |
| 5. I tend not to notice feelings of physical tension or discomfort until they really grab my attention | 0.31 | 0.17 | -0.08 |
| 6. I forget a person's name almost as soon as I've been told it for the first time. | -0.14 | 0.25 | -0.12 |
| 7. It seems I'm "running on automatic" without much awareness of what I'm doing. | -0.11 | 0.05 | -0.08 |
| 8. I rush through activities without being really attentive to them. | 0.13 | -0.02 | -0.12 |
| 9. I get so focused on the goal I want to achieve that I lost touch with what I am doing right now to get there | 0.01 | -0.10 | -0.12 |
| 10. I do jobs or tasks automatically, without being aware of what I'm doing | 0.01 | 0.06 | -0.26 |
| 11. I find myself listening to someone with one ear, doing something else at the same time. | 0.14 | 0.33 | 0.13 |
| 12. I drive places on "automatic pilot" and then wonder why I went there. | -0.34 | -0.22 | 0.29 |
| 13. I find myself preoccupied with the future or the past | -0.24 | -0.21 | 0.07 |
| 14. I find myself doing things without paying attention. | 0.14 | -0.14 | 0.08 |
| 15. I snack without being aware that I'm eating. | 0.11 | -0.08 | 0.13 |

DIF contrasts = Difficulty in subgroup 1 –difficulty in subgroup 2, and all DIF contrasts were nonsignificant.

[a] assessed by Insomnia Severity Index with a cutoff score of 15 (suggesting moderate to severe insomnia)

[b] assessed by 12-item general health questionnaire (GHQ-12) with scores of 12

[c] patients were classified into older (higher than mean age >39.44) and younger (lower than mean age ≤39.44)

construct of mindfulness. A possible reason for the different findings may be due to the different language versions: Goh et al. [47] examined the English MAAS and found 5 misfit items; Inchausti et al. [20] examined the Spanish MAAS and found 2 misfit items; we examined the Persian MAAS and found no misfit items. Another possible reason for the different findings may be due to the different models used in the Rasch: Goh et al. [47] used partial credit model; Inchausti et al. [20] and we used rating scale model. However, future studies are warranted to accumulate evidence of Rasch analysis results on the MASS because such evidence is little among current literature.

Because prisoners usually encounter substantial distress due to several imprisonment factors [27], they may need to receive appropriate interventions to prevent or to treat their distress. In this regard, MBI is a potential approach for healthcare providers to help the prisoners. With the verified psychometric properties of the MAAS, healthcare providers can use the MAAS to monitor the effectiveness of the MBI designed for prisoners. However, cautious should be paid attention to because the content validity of the MAAS only contributes to the mindlessness facet of mindfulness. Therefore, healthcare providers may want to integrate the MAAS with other mindfulness instrument to evaluate the effects of mindful programs.

There are some limitations in this study. First, only male prisoners were recruited in this study. Therefore, it is unclear whether the promising psychometric properties of MAAS found in this study can be generalized to female prisoners given that males and females have different responses to psychological distress. Second, the male prisoners were recruited from the same institution; therefore, the generalizability of our findings may not extend to other settings.

Third, only construct validity of the MAAS was assessed in the present study; therefore, other important psychometric properties such as reproducibility (i.e., test-retest reliability) and responsiveness (i.e., whether the MAAS can effectively detect mindfulness improvement) remain unknown. Future studies may thus want to assess other properties of the MAAS among prisoners. Fourth, all the instruments were self-reported; therefore, the commonly encountered biases (e.g., recall bias and social desirability) were hard to control in this study. However, given that all the self-reported measures used in the present study have good psychometric properties, the bias problems might not be serious.

## Conclusion

In conclusion, our findings indicated that the Persian MAAS had robust properties in its construct validity when assessing male prisoners. Specifically, two test theories have been applied and both theories supported the construct validity of the Persian MAAS. The scale may be used for prisoners to detect and monitor their mindfulness condition. For example, the Persian MAAS can be used to examine whether any MBIs designed for prisoners have intervention effects.

## Supporting information

**S1 Data.**
(XLSX)

## Author Contributions

**Conceptualization:** Ali Poorebrahim, Chung-Ying Lin, Amir H. Pakpour.

**Data curation:** Ali Poorebrahim, Vida Imani, Narges Ehsani.

**Formal analysis:** Amir H. Pakpour.

**Investigation:** Ali Poorebrahim, Vida Imani, Shapour Soltankhah Kolvani, Seyed Abbas Alaviyoun, Amir H. Pakpour.

**Methodology:** Chung-Ying Lin, Amir H. Pakpour.

**Project administration:** Ali Poorebrahim, Vida Imani, Shapour Soltankhah Kolvani, Seyed Abbas Alaviyoun.

**Resources:** Amir H. Pakpour.

**Supervision:** Amir H. Pakpour.

**Validation:** Chung-Ying Lin, Vida Imani, Shapour Soltankhah Kolvani, Seyed Abbas Alaviyoun, Amir H. Pakpour.

**Visualization:** Amir H. Pakpour.

**Writing – original draft:** Chung-Ying Lin.

**Writing – review & editing:** Ali Poorebrahim, Chung-Ying Lin, Vida Imani, Shapour Soltankhah Kolvani, Seyed Abbas Alaviyoun, Narges Ehsani, Amir H. Pakpour.

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
