## [Editor Report · Decision Letter 0]

6 Apr 2020

PONE-D-20-09274

Using Mindful Attention Awareness Scale on male prisoners: confirmatory factor analysis and Rasch models

PLOS ONE

Dear Dr. Pakpour,

Thank you for submitting your manuscript to PLOS ONE. After careful consideration, we have decided that your manuscript does not meet our criteria for publication and must therefore be rejected.

Specifically:

The methods are unclear. Did you use CFA for oridnal indicators (items) as implemented in Mplus or the R-package lavaan ?

The results are lacking. You should report chi-square, df and P-value for the CFA fit. You should also report the RMSEA and corresponding confidence interval. I would be appropritae to cite literature indicating the type I error rates associated with the Rasch item fit statistics. You do not report on local response dependence - this is unacceptable.

How can you state that 'CFA results showed a single factor solution for the Persian MAAS', when CFI=0.928 and TLI=0.908 ?

I am sorry that we cannot be more positive on this occasion, but hope that you appreciate the reasons for this decision. I also hope that my comments can help you revise the manuscript if you decide to submit it elsewhere.

Yours sincerely,

Karl Bang Christensen, Ph.D.

Academic Editor

PLOS ONE

Additional Editor Comments (if provided):

The methods are unclear. Did you use CFA for oridnal indicators (items) as implemented in Mplus or the R-package lavaan ?

The results are lacking. You should report chi-square, df and P-value for the CFA fit. You should also report the RMSEA and corresponding confidence interval. I would be appropritae to cite literature indicating the type I error rates associated with the Rasch item fit statistics. You do not report on local response dependence - this is unacceptable.

How can you state that 'CFA results showed a single factor solution for the Persian MAAS', when CFI=0.928 and TLI=0.908 ?

- - - - -

---

## [Author Response · Author response to Decision Letter 0]

21 Apr 2020

Dear Editor-in-Chief, 

Thank you for inviting us to revise our manuscript entitled “Using Mindful Attention Awareness Scale on male prisoners: confirmatory factor analysis and Rasch models” (PONE-D-20-09274).

Below we have provided our point-by-point reply to the comments made by the previous academic editor. 

We look forward to your further comments.

Sincerely yours,

Amir H. Pakpour, PhD

Chung-Ying Lin, PhD

Response to Comments: 

1. The methods are unclear. Did you use CFA for ordinal indicators (items) as implemented in Mplus or the R-package lavaan ?

Reply: We have now provided the software information in the revised. 

“Apart from the descriptive statistics (mean and SD for continuous variables; frequency and percentage for categorical variables), confirmatory factor analysis (CFA) using AMOS 24.0 and Rasch analysis using WINSTEPS 4.3.0 (winsteps.com) were used to test the construct validity of the Persian MAAS among male prisoners.”

2. The results are lacking. You should report chi-square, df and P-value for the CFA fit. You should also report the RMSEA and corresponding confidence interval. I would be appropriate to cite literature indicating the type I error rates associated with the Rasch item fit statistics. You do not report on local response dependence - this is unacceptable.

Reply: We have now clearly stated the information.

“The one-factor structure of the MAAS is verified by the fit indices of the CFA, including CFI (0.928), TLI (0.908), RMSEA (0.063), 95% CI of RMSEA (0.052, 0.074), and SRMR (0.049), except for the significant χ2 test (Table 2).”

“Rasch analysis further shows that The global test on type I error rates associated with the Rasch item fit statistics was nonsignificant (p = 0.48), indicating the satisfactory item fit across all the items.”

“Tests on local response dependence indicate that there were no substantial residual correlations among the items (r = -0.33 to 0.27).”

3. How can you state that 'CFA results showed a single factor solution for the Persian MAAS', when CFI=0.928 and TLI=0.908 ?

Reply: The practice of CFI and TLI > 0.9 to indicate data-model fit is widely used in the literature, including those published recently. Also, statistics textbook also recommends the cutoff at 0.9. Therefore, we are confident that using CFI = 0.928 and TLI = 0.908 (together with other CFA fit indices) can support the decision of a single factor solution for the Persian MAAS. 

Ref: 

Hoyle, R. H., & Panter, A. T. (1995). Writing about structural equation modeling. In R. H. Hoyle (Ed.), Structural equation modeling: Concepts, issues, and application. Thousand Oaks, CA: Sage. 

Kline, R. B. (1998). Principles and practice of structural equation modeling. New York: Guilford Press.

Leung, H., Pakpour, A. H., Strong, C., Lin, Y.-C., Tsai, M.-C., Griffiths, M. D., Lin, C.-Y., Chen, I.-H. (2020). Measurement invariance across young adults from Hong Kong and Taiwan among three internet-related addiction scales: Bergen Social Media Addiction Scale (BSMAS), Smartphone Application-Based Addiction Scale (SABAS), and Internet Gaming Disorder Scale-Short Form (IGDS-SF9)(Study Part A). Addictive Behaviors, 101, 105969.

Chen, I.-H., Strong, C., Lin, Y.-C., Tsai, M.-C., Leung, H., Lin, C.-Y., Pakpour, A. H., Griffiths, M. D. (2020). Time invariance of three ultra-brief internet-related instruments: Smartphone Application-Based Addiction Scale (SABAS), Bergen Social Media Addiction Scale (BSMAS), and the nine-item Internet Gaming Disorder Scale- Short Form (IGDS-SF9) (Study Part B). Addictive Behaviors, 101, 105960.

Pakpour, A. H., Tsai, M.-C., Lin, Y.-C., Strong, C., Latner, J. D., Fung, X. C. C., Lin, C.-Y., & Tsang, H. W. H. (2019). Psychometric properties and measurement invariance of the Weight Self-Stigma Questionnaire and Weight Bias Internalization Scale in Hongkongese children and adolescents. International Journal of Clinical and Health Psychology, 19, 150-159.

---

## [Decision Letter · Decision Letter 1]

4 Feb 2021

PONE-D-20-09274R1

Using Mindful Attention Awareness Scale on male prisoners: confirmatory factor analysis and Rasch models

PLOS ONE

Dear Dr. Pakpour,

Thank you for submitting your manuscript to PLOS ONE. After careful consideration, we feel that it has merit but does not fully meet PLOS ONE’s publication criteria as it currently stands. Therefore, we invite you to submit a revised version of the manuscript that addresses the points raised during the review process.

Please accept our sincere apologies for the time it has taken to process the appeal decision on your manuscript. Unfortunately, we were not able to secure an academic editor to assess your manuscript and this has resulted in delays in the processing of your manuscript.

Your manuscript has been evaluated by 3 expert reviewers and you will find their comments below. The reviews offer their praise for several elements of the manuscript, including the importance of studying the sample population and the sample size employed.

Please note that as per our publication criteria, PLOS ONE requires that all experiments, statistics and other analyses are performed to a high technical standard, described in sufficient detail and adhere to appropriate reporting guidelines and community standards. Conclusions must be presented in an appropriate fashion and be supported by the data (Please see http://journals.plos.org/plosone/s/criteria-for-publication). The reviewers’ comments concern the statistical analysis performed, including the justification for the methods employed and the ability of the results to support the conclusions.

One of the reviewers has also provided comments encouraging you to engage in open science practices. All PLOS journals require authors to make all data necessary to replicate their study’s findings publicly available without restriction at the time of publication. When specific legal or ethical restrictions prohibit public sharing of a data set, authors must indicate how others may obtain access to the data. Authors must share the “minimal data set” for their submission. PLOS defines the minimal data set to consist of the data required to replicate all study findings reported in the article, as well as related metadata and methods. (Please see here: https://journals.plos.org/plosone/s/data-availability).

I note that you have indicated that all data are fully available without restriction all relevant data are within the manuscript and its Supporting Information files. However, no Supporting Information files have been included. In your revised manuscript please ensure that you provide details of where your data are deposited and how these can be accessed. More information on data deposition methods is provided in the link provided in the previous paragraph. Whilst only data sharing is a requirement for publication, I do encourage to make your scripts available to further improve the reproducibility of your work. If you provide any materials used in your study as Supporting Information files please ensure that you have permission to reproduce these with a CC BY license (please see here for more details: https://journals.plos.org/plosone/s/licenses-and-copyright).

We look forward to receiving your revised manuscript. Please accept my apologies again for the delay in processing your submission.

We look forward to receiving your revised manuscript.

Kind regards,

George Vousden

Senior Staff Editor

PLOS ONE

Journal Requirements:

2. Please describe what considerations were made for the prisoners included in this study. For instance, were participants able to opt out of the study? Did individuals who did not participate receive the same treatment offered to participants?

4. Thank you for updating your data availability statement. You note that your data are available within the Supporting Information files, but no such files have been included with your submission. At this time we ask that you please upload your minimal data set as a Supporting Information file, or to a public repository such as Figshare or Dryad.

Please also ensure that when you upload your file you include separate captions for your supplementary files at the end of your manuscript.

As soon as you confirm the location of the data underlying your findings, we will be able to proceed with the review of your submission.

5. If possible, please upload a file showing your changes either highlighted or using track changes. This should be uploaded as a Revised Manuscript w/tracked changes, file type. Please follow this link for more information: http://blogs.PLOS.org/everyone/2011/05/10/how-to-submit-your-revised-manuscript/

Reviewers' comments:

Reviewer's Responses to Questions

**Comments to the Author**

1. If the authors have adequately addressed your comments raised in a previous round of review and you feel that this manuscript is now acceptable for publication, you may indicate that here to bypass the “Comments to the Author” section, enter your conflict of interest statement in the “Confidential to Editor” section, and submit your "Accept" recommendation.

Reviewer #1: All comments have been addressed

Reviewer #2: All comments have been addressed

Reviewer #3: (No Response)

2. Is the manuscript technically sound, and do the data support the conclusions?

Reviewer #1: Yes

Reviewer #2: Partly

Reviewer #3: Yes

3. Has the statistical analysis been performed appropriately and rigorously? 

Reviewer #1: Yes

Reviewer #2: No

Reviewer #3: Yes

4. Have the authors made all data underlying the findings in their manuscript fully available?

Reviewer #1: Yes

Reviewer #2: No

Reviewer #3: Yes

5. Is the manuscript presented in an intelligible fashion and written in standard English?

Reviewer #1: Yes

Reviewer #2: Yes

Reviewer #3: Yes

6. Review Comments to the Author

Reviewer #1: After reviewing the new manuscript and the responses to the reviewers, I consider that this version meets all the requirements to be published in Plos One as an original article.

I have no further comment to make.

Reviewer #2: Overall, I see added value on this paper. However, there are several necessary changes (especially regarding the introduction and theoretical sections) that need to occur prior to acceptance. I provide a list below. Perhaps the best added value of this contribution is the access to a big hard-to-reach sample of prisoners from a non-WEIRD population. A focus on this would improve the paper significantly.

Abstract:

-Fit indices are not required in the abstract, consider removing them.

Introduction:

-The whole introduction needs restructuring. While I see added value on validating the MAAS in this population and I see worthy work in the authors' efforts, the core messages of the introduction are misguided. The MAAS is right now an outdated instrument for Western scenarios (e.g. integration in other instruments, and content validity issues since it measures Mindlessness or Automatic Pilot, not mindfulness), used in laboratory settings due to its simplicity. However, it is important to provide validated instruments world-wide, so I see value in this paper. A restructured introduction in this direction would be a great (and necessary) plus for the paper (and accordingly, changes in discussion). Some other inquiries are provided below.

-MBIs count with heavier evidence than those provided. Consider adding reivews or meta-analyses when picturing evidence on MBIs (e.g. Creswell, 2017, Annual review of Psychology).

-Current evidence on mindfulness models rely strongly on multidimensional definitions (e.g. Lindsay & Creswell, 2017; Hölzel et al, 2011).

-Psychometric developments on mindfulness state clearly that the MAAS has been already integrated in other multidimensional instruments (e.g. FFMQ).

Methods:

- I see of great value to provide such a big sample for a hard-to-reach scneario such as prisons. My congratulations.

- Instruments are provided with Cronbach's alpha. Although this is widely used, is flawed (see McNeish, 2018, Viladrich, Angulo-Brunet & Doval, 2017, and Trizano-Hermosilla & Alvarado, 2016, for references). Given all self-reports are ordinal and probably with normality or tau-equivalence issues, I recommend to remove alpha and provide mcdonald's omega, guttman's lambda6, GLB or other similar indices. The JASP software is a free and effective tool to obtain these indices (download at: https://jasp-stats.org/download/)

- CFA was used with maximum likelihood when ordinal items were implemented. This is open to method bias since ML assumes items to be continuous and normal. Current estimation methods for ordinal items are (1) Weighted Least Squares Mean-and-Variance Adjusted (WLSMV) or Maximum Likelihood Robust (MLR), and (2) polychoric correlations. If the authors encounter software limitations, MPlus and lavaan package of R provide all these requirements.

- Why Rasch models? There is no justification on why these models are better than other IRT models (e.g. Likelihood Ratio Tests for 1 (Rasch), 2, 3 or 4 paremeter logistics models). Finally, why rating scale model? Why not, for example, the graded response model? These choices need to be justified. In addition, local independence check methods need to be specified in the Data Analysis section.

- It would be interesting to test external validity of the MAAS in this sample beyond DIF. For example, correlation analysis for the MAAS scores with the ISI and GHQ scores.

- Psychology is right now facing a reproducibility crisis (see Munafó et al, 2017, for details). Is there a possibility for the authors engaging in open science practices? For example, sharing openly in the Open Science Framework the data, scipts, and other materials to ensure reproduction of results. Since the data may encounter privacy issues due to prison environment, this is not a request but a suggestion for general improvement of psychological science. In any case, I did not see any statement on this matter in the manuscript.

Discussion:

- "Our Rasch analysis results corresponded to our CFA results; that is, both psychometric

testing supported the one-factor structure of the MAAS and no items were misfit". This is relatively obvious since the same sample was applied. Consider removing.

- The practical implications need to consider the content validity issues of the MAAS (measuring the mindlessness facet of mindfulness).

Reviewer #3: Introduction:

Page 3: Reference to mediation in second paragraph. I'm not clear on why the general population's experience of meditation relates to the factor structure of mindfulness as a construct.

Page 4, first paragraph: "brief state" versions - do you mean brief versions, or versions of the MAAS intended to measure state vs trait mindfulness?

Results:

The term "verified" is used to describe a one-factor structure, this wording is too strong. First, though many of the fit statistics are acceptable, the RMSEA is outside of the good fit range, and these statistics are often most useful in comparison to other models (i.e., there could be a 3 factors structure with even better fit statistics). Are there are factor structures of the MAAS that have been proposed or found (such as the Iranian study cited in the Discussion)? If so, please test in comparison. If not, please just change wording to "one-factor structure is supported by the fit indices"

Also, I think it would be valuable for this study to report descriptive statistics of the MAAS and compare the values in this population to others in other samples.

7. PLOS authors have the option to publish the peer review history of their article (what does this mean?). If published, this will include your full peer review and any attached files.

Reviewer #1: No

Reviewer #2: **Yes: **Oscar Lecuona

Reviewer #3: No

---

## [Author Response · Author response to Decision Letter 1]

10 Feb 2021

Dear Dr. Vousden, 

Thank you for inviting us to revise our manuscript entitled “Using Mindful Attention Awareness Scale on male prisoners: confirmatory factor analysis and Rasch models” (PONE-D-20-09274R1).

Below we have provided our point-by-point reply to the comments made by the three reviewers. All the revisions are presented using red fonts in the revised manuscript. We deeply appreciate their comments, which help us substantially improve our work. 

We look forward to your further comments.

Sincerely yours,

Amir H. Pakpour, PhD

Chung-Ying Lin, PhD

Response to Editor: 

1. Please note that as per our publication criteria, PLOS ONE requires that all experiments, statistics and other analyses are performed to a high technical standard, described in sufficient detail and adhere to appropriate reporting guidelines and community standards. Conclusions must be presented in an appropriate fashion and be supported by the data (Please see http://journals.plos.org/plosone/s/criteria-for-publication). The reviewers’ comments concern the statistical analysis performed, including the justification for the methods employed and the ability of the results to support the conclusions.

Reply: We have now clearly responded to the reviewers’ comments regarding the justifications of choosing specific statistical methods. Please see our detailed response to Reviewer #1’s comment 4.

2. One of the reviewers has also provided comments encouraging you to engage in open science practices. All PLOS journals require authors to make all data necessary to replicate their study’s findings publicly available without restriction at the time of publication. When specific legal or ethical restrictions prohibit public sharing of a data set, authors must indicate how others may obtain access to the data. Authors must share the “minimal data set” for their submission. PLOS defines the minimal data set to consist of the data required to replicate all study findings reported in the article, as well as related metadata and methods. (Please see here: https://journals.plos.org/plosone/s/data-availability).

I note that you have indicated that all data are fully available without restriction all relevant data are within the manuscript and its Supporting Information files. However, no Supporting Information files have been included. In your revised manuscript please ensure that you provide details of where your data are deposited and how these can be accessed. More information on data deposition methods is provided in the link provided in the previous paragraph. Whilst only data sharing is a requirement for publication, I do encourage to make your scripts available to further improve the reproducibility of your work. If you provide any materials used in your study as Supporting Information files please ensure that you have permission to reproduce these with a CC BY license (please see here for more details: https://journals.plos.org/plosone/s/licenses-and-copyright).

Reply: We have now added the dataset but we believe that our data is partly sensitive because they are prisoners. We need to make sure from our institution if we can publish our whole dataset. 

3. Please describe what considerations were made for the prisoners included in this study. For instance, were participants able to opt out of the study? Did individuals who did not participate receive the same treatment offered to participants?

Reply: Thanks for your comment. The participation to the study was voluntary and anonymous. There were no any interventional materials (or treatment) as this study was a cross-sectional study. The participants were only asked to complete the study measure after describing study aims and checking eligibility criteria. 

Response to Reviewer #1: 

1. After reviewing the new manuscript and the responses to the reviewers, I consider that this version meets all the requirements to be published in Plos One as an original article. I have no further comment to make.

Reply: Thank you for the positive comment.

Response to Reviewer #2: 

1. Overall, I see added value on this paper. However, there are several necessary changes (especially regarding the introduction and theoretical sections) that need to occur prior to acceptance. I provide a list below. Perhaps the best added value of this contribution is the access to a big hard-to-reach sample of prisoners from a non-WEIRD population. A focus on this would improve the paper significantly.

Reply: Thank you for the positive comment. We also appreciate your following comments, which help us substantially improve this work. 

2. Abstract:

-Fit indices are not required in the abstract, consider removing them.

Reply: We have now removed the fit indices from the Abstract.

“The CFA results showed a single factor solution for the Persian MAAS.”

3. Introduction:

-The whole introduction needs restructuring. While I see added value on validating the MAAS in this population and I see worthy work in the authors' efforts, the core messages of the introduction are misguided. The MAAS is right now an outdated instrument for Western scenarios (e.g. integration in other instruments, and content validity issues since it measures Mindlessness or Automatic Pilot, not mindfulness), used in laboratory settings due to its simplicity. However, it is important to provide validated instruments world-wide, so I see value in this paper. A restructured introduction in this direction would be a great (and necessary) plus for the paper (and accordingly, changes in discussion). Some other inquiries are provided below.

-MBIs count with heavier evidence than those provided. Consider adding reviews or meta-analyses when picturing evidence on MBIs (e.g. Creswell, 2017, Annual review of Psychology).

-Current evidence on mindfulness models rely strongly on multidimensional definitions (e.g. Lindsay & Creswell, 2017; Hölzel et al, 2011).

-Psychometric developments on mindfulness state clearly that the MAAS has been already integrated in other multidimensional instruments (e.g. FFMQ).

Reply: Thank you for the guidance in the Introduction and also the valuable references. We have now restructured the Introduction with the incorporation of your suggested references. 

“Moreover, a review summarized the effectiveness of MBIs through evaluating randomized controlled trials (RCTs) and found that the analyzed rigorous RCTs showing the effectiveness of MBIs on different outcomes, including addition, chronic pain, and depression relapse [12].”

“The concept of “mindfulness” has been argued for its operational definition [13] because of the arguments on whether mindfulness is a single factor [1,14,15] or a multifactor [5,16] construct. However, current evidence on mindfulness models is prone to multifactor [17,18].”

“Although the MAAS suffers from some problems in its content validity (i.e., only assessing part of the mindfulness via mindlessness facet) and becomes outdated in the Western country, it deserves a breaking point for the countries without appropriate instruments assessing mindfulness.”

“In addition, the MAAS has been integrated into other mindfulness instruments to assess the nature of multifactor structure in mindfulness (e.g., the Five Facet Mindfulness Questionnaire) [26].”

References: 

12. Creswell JD. Mindfulness interventions. Annu Rev Psychol. 2017;68(1):491-516.

17. Hölzel BK, Carmody J, Vangel M, et al. Mindfulness practice leads to increases in regional brain gray matter density. Psychiatry Res. 2011;191(1):36-43. doi:10.1016/j.pscychresns.2010.08.006

18. Lindsay EK, Creswell JD. Mechanisms of mindfulness training: Monitor and Acceptance Theory (MAT). Clin Psychol Rev. 2017;51:48-59. doi:10.1016/j.cpr.2016.10.011

26. Baer RA, Smith GT, Hopkins J, Krietemeyer J, Toney L. Using self-report assessment methods to explore facets of mindfulness. Assessment. 2006;13(1):27-45. doi:10.1177/1073191105283504

4. Methods:

- I see of great value to provide such a big sample for a hard-to-reach scenario such as prisons. My congratulations.

- Instruments are provided with Cronbach's alpha. Although this is widely used, is flawed (see McNeish, 2018, Viladrich, Angulo-Brunet & Doval, 2017, and Trizano-Hermosilla & Alvarado, 2016, for references). Given all self-reports are ordinal and probably with normality or tau-equivalence issues, I recommend to remove alpha and provide mcdonald's omega, guttman's lambda6, GLB or other similar indices. The JASP software is a free and effective tool to obtain these indices (download at: https://jasp-stats.org/download/)

- CFA was used with maximum likelihood when ordinal items were implemented. This is open to method bias since ML assumes items to be continuous and normal. Current estimation methods for ordinal items are (1) Weighted Least Squares Mean-and-Variance Adjusted (WLSMV) or Maximum Likelihood Robust (MLR), and (2) polychoric correlations. If the authors encounter software limitations, MPlus and lavaan package of R provide all these requirements.

- Why Rasch models? There is no justification on why these models are better than other IRT models (e.g. Likelihood Ratio Tests for 1 (Rasch), 2, 3 or 4 paremeter logistics models). Finally, why rating scale model? Why not, for example, the graded response model? These choices need to be justified. In addition, local independence check methods need to be specified in the Data Analysis section.

- It would be interesting to test external validity of the MAAS in this sample beyond DIF. For example, correlation analysis for the MAAS scores with the ISI and GHQ scores.

- Psychology is right now facing a reproducibility crisis (see Munafó et al, 2017, for details). Is there a possibility for the authors engaging in open science practices? For example, sharing openly in the Open Science Framework the data, scipts, and other materials to ensure reproduction of results. Since the data may encounter privacy issues due to prison environment, this is not a request but a suggestion for general improvement of psychological science. In any case, I did not see any statement on this matter in the manuscript.

Reply: Thank you. We have now made the following revisions in the Methods and Results.

(1) Both Cronbach’s alpha and McDonald’s omega are presented for the used instruments.

“Moreover, the internal consistency of the MAAS was adequate (Cronbach’s α=0.878; McDonalds’ ω=0.879).”

“Moreover, the internal consistency of the ISI was adequate (Cronbach’s α=0.900; McDonalds’ ω=0.902).”

“Moreover, the internal consistency of the GHQ-12 was adequate (Cronbach’s α=0.714; McDonalds’ ω=0.653).”

(2) We have now used weighted least squares mean-and-variance adjusted (WLSMV) estimator in the CFA. The results are revised accordingly (Please refer to Tables 2 and 3).

“In the CFA, a one-factor structure was tested for the MAAS using the diagonally weighted least squares (DWLS) estimator.”

“The one-factor structure of the MAAS is supported by the fit indices of the CFA, including CFI (0.982), TLI (0.979), RMSEA (0.044), 95% CI of RMSEA (0.032, 0.055), and SRMR (0.064), except for the significant χ2 test (Table 2). Moreover, all the factor loadings in the MAAS were strong (0.478 to 0.679) and significant (Table 3).”

(3) We have now provided the justifications regarding the use of Rasch model and rating scale model. The local independence check methods are clearly mentioned in the Data Analysis section.

“We considered using Rasch model with RSM instead of other item-response theory models (e.g., the two parameter or the three logistic parameter model with partial credit model or graded response model) because Rasch model with RSM can provide simpler estimation in modeling (i.e., Rasch model with RSM needs not to estimate other parameter like discrimination and the category difference for every two responses). Thus, the benefit of using such a model than other types of item-response theory model is it fits better with the principle of parsimony.”

“Local independence was tested to understand whether residual correlations exist among items. Specifically, we tested the Rasch residual for every item and used the correlations between the Rasch residuals to examine the local independence, where an absolute correlation coefficient > 0.4 indicating substantial dependence.”

(4) The correlations between the MAAS, ISI, and GHQ scores are conducted and presented.

“Apart from the CFA and Rasch, Pearson correlations were carried to understand the associations between the MAAS, the ISI, and the GHQ-12 scores.”

“Moreover, the MAAS total score was significantly correlated with the ISI (r=-0.60; p<0.001) and the GHQ-12 (r=-0.44; p<0.001) scores.”

(5) We have now added the dataset but we believe that our data is partly sensitive because they are prisoners. We need to make sure from our institution if we can publish our whole dataset. 

5. Discussion:

- "Our Rasch analysis results corresponded to our CFA results; that is, both psychometric testing supported the one-factor structure of the MAAS and no items were misfit". This is relatively obvious since the same sample was applied. Consider removing.

- The practical implications need to consider the content validity issues of the MAAS (measuring the mindlessness facet of mindfulness).

Reply: We have now removed the sentence “Our Rasch analysis results corresponded to our CFA results; that is, both psychometric testing supported the one-factor structure of the MAAS and no items were misfit”. Also, we have mentioned the content validity issues of the MAAS in the practical implications.

“However, cautious should be paid attention to because the content validity of the MAAS only contributes to the mindlessness facet of mindfulness. Therefore, healthcare providers may want to integrate the MAAS with other mindfulness instrument to evaluate the effects of mindful programs.”

Response to Reviewer #3: 

1. Page 3: Reference to mediation in second paragraph. I'm not clear on why the general population's experience of meditation relates to the factor structure of mindfulness as a construct.

Reply: We have now revised the descriptions here. 

“Nevertheless, a single factor (i.e., attention and awareness of the present) can be used as the basis to assess mindfulness before expanding the concept of mindfulness to pose various dimensionalities, considering that a single-factor structure is easier to be studied and understood than a multi-factor structure.”

2. Page 4, first paragraph: "brief state" versions - do you mean brief versions, or versions of the MAAS intended to measure state vs trait mindfulness?

Reply: They are brief versions. We have now corrected the descriptions.

“Additionally, several brief versions of the MAAS have been developed and validated (e.g., [21,22])”

3. Results:

The term "verified" is used to describe a one-factor structure, this wording is too strong. First, though many of the fit statistics are acceptable, the RMSEA is outside of the good fit range, and these statistics are often most useful in comparison to other models (i.e., there could be a 3 factors structure with even better fit statistics). Are there are factor structures of the MAAS that have been proposed or found (such as the Iranian study cited in the Discussion)? If so, please test in comparison. If not, please just change wording to "one-factor structure is supported by the fit indices"

Reply: Thank you for the guidance. We have now revised the wording to “supported”. 

“The one-factor structure of the MAAS is supported by the fit indices of the CFA, including CFI (0.928), TLI (0.908), RMSEA (0.063), 95% CI of RMSEA (0.052, 0.074), and SRMR (0.049), except for the significant χ2 test (Table 2)”

4. Also, I think it would be valuable for this study to report descriptive statistics of the MAAS and compare the values in this population to others in other samples.

Reply: We have now described the descriptive statistics and compared with other studies.

“On average, the MAAS mean score was 4.21 (0.91).”

“Moreover, the MAAS mean score found in the present sample of prisoners was slightly higher than that of college students (MAAS mean score=3.89 for Bangla students [19]; 3.88 for Greek students [44]; and 3.72 to 4.01 for American students [1,30,45]) but lower than that of general population (MAAS mean score=4.86 for Italians [23] and 4.45 for Canadians [46]).”

---

## [Decision Letter · Decision Letter 2]

4 May 2021

PONE-D-20-09274R2

Using Mindful Attention Awareness Scale on male prisoners: confirmatory factor analysis and Rasch models

PLOS ONE

Dear Dr. Pakpour,

Thank you for submitting your manuscript to PLOS ONE. The revision was adequate, there is a minor issue left. PLease see the comments of a single reviewer. We invite you to submit a revised version of the manuscript that addresses the single point raised during the review process.

We look forward to receiving your revised manuscript.

Kind regards,

Gilles van Luijtelaar, Ph.D.

Academic Editor

PLOS ONE

Journal Requirements:

Additional Editor Comments (if provided):

There is a minor issue left, as can be seen from the comments of a single reviewer concerning the use of pearson's correlation coefficient and whether the assumptions for the usage of this paramentric statistic is allowed.

Next two typo's. Otherwise it is fine.

Reviewers' comments:

Reviewer's Responses to Questions

**Comments to the Author**

1. If the authors have adequately addressed your comments raised in a previous round of review and you feel that this manuscript is now acceptable for publication, you may indicate that here to bypass the “Comments to the Author” section, enter your conflict of interest statement in the “Confidential to Editor” section, and submit your "Accept" recommendation.

Reviewer #2: All comments have been addressed

Reviewer #3: All comments have been addressed

2. Is the manuscript technically sound, and do the data support the conclusions?

Reviewer #2: Yes

Reviewer #3: Yes

3. Has the statistical analysis been performed appropriately and rigorously? 

Reviewer #2: Yes

Reviewer #3: Yes

4. Have the authors made all data underlying the findings in their manuscript fully available?

Reviewer #2: Yes

Reviewer #3: Yes

5. Is the manuscript presented in an intelligible fashion and written in standard English?

Reviewer #2: Yes

Reviewer #3: Yes

6. Review Comments to the Author

Reviewer #2: Thanks to the authors for addressing all the comments. I have a relevant comment and some minor corrections. Apart from that, I have no further comment on the manuscript.

- There is no justification on why estimating correlations between instrument scores with Pearson correlations. Assumption checks for normality and outliers is necessary to ensure Pearson's r estimates the correlations adequately, and if not, use alternative estimators (e.g. Spearman's rho or Kendall's tau-b).

- "Western country" instead of "Western countries" (p. 5).

- "Modern test theory" instead of "item response theory" (p. 6).

Reviewer #3: All issues addressed, thank you! I have no further issues and am recommending publication of this manuscript.

7. PLOS authors have the option to publish the peer review history of their article (what does this mean?). If published, this will include your full peer review and any attached files.

Reviewer #2: **Yes: **Oscar Lecuona

Reviewer #3: No

---

## [Author Response · Author response to Decision Letter 2]

10 May 2021

Dear Dr. Vousden, 

Thank you for inviting us to revise our manuscript entitled “Using Mindful Attention Awareness Scale on male prisoners: confirmatory factor analysis and Rasch models” (PONE-D-20-09274R2).

Below we have provided our point-by-point reply to the comments made by the three reviewers. All the revisions are presented using red fonts in the revised manuscript. We deeply appreciate their comments, which help us substantially improve our work. 

We look forward to your further comments.

Sincerely yours,

Amir H. Pakpour, PhD

Chung-Ying Lin, PhD

Response to Reviewer #2: 

1. - There is no justification on why estimating correlations between instrument scores with Pearson correlations. Assumption checks for normality and outliers is necessary to ensure Pearson's r estimates the correlations adequately, and if not, use alternative estimators (e.g. Spearman's rho or Kendall's tau-b).

Reply: We have conducted Spearman's rho correlations for the above mentioned associations. 

2. - "Western country" instead of "Western countries" (p. 5).

- "Modern test theory" instead of "item response theory" (p. 6).

Reply: We have now edited these. Thanks

---

## [Editor Report · Decision Letter 3]

26 May 2021

PONE-D-20-09274R3

Using Mindful Attention Awareness Scale on male prisoners: confirmatory factor analysis and Rasch models

PLOS ONE

Dear Dr. Pakpour,

Thank you for submitting your manuscript to PLOS ONE. A very small issue remains. Therefore, we invite you to submit a revised version of the manuscript that addresses the points raised during the review process.

We look forward to receiving your revised manuscript.

Kind regards,

Gilles van Luijtelaar, Ph.D.

Academic Editor

PLOS ONE

Journal Requirements:

Additional Editor Comments (if provided):

Please add a rationale, after checking for outliers and normality of the distributions, for using a parametric correlation coefficient, or use a non-parametric one to be on the safe side.

The reviewer was happy with all the changes made by you. A minor issue that the reviewer noticed was that there was no justification on why estimating correlations between instrument scores with Pearson correlations. Assumption checks for normality and outliers are necessary to ensure Pearson's r estimates the correlations adequately, and if not, use alternative estimators (e.g. Spearman's rho or Kendall's tau-b).

- "Western country" instead of "Western countries" (p. 5).

- "Modern test theory" instead of "item response theory" (p. 6).

---

## [Author Response · Author response to Decision Letter 3]

8 Jun 2021

Dear Prof. Dr. van Luijtelaar, 

Thank you for inviting us to revise our manuscript entitled “Using Mindful Attention Awareness Scale on male prisoners: confirmatory factor analysis and Rasch models” (PONE-D-20-09274R3).

Below we have provided our point-by-point reply to the comments made by the three reviewers. All the revisions are presented using red fonts in the revised manuscript. We deeply appreciate their comments, which help us substantially improve our work. 

We look forward to your further comments.

Sincerely yours,

Amir H. Pakpour, PhD

Chung-Ying Lin, PhD

Response to editor: 

1. - Please add a rationale, after checking for outliers and normality of the distributions, for using a parametric correlation coefficient, or use a non-parametric one to be on the safe side.

The reviewer was happy with all the changes made by you. A minor issue that the reviewer noticed was that there was no justification on why estimating correlations between instrument scores with Pearson correlations. Assumption checks for normality and outliers are necessary to ensure Pearson's r estimates the correlations adequately, and if not, use alternative estimators (e.g. Spearman's rho or Kendall's tau-b).

Reply: Thank you for the positive feedback. We agree that using nonparametric estimator for the correlations is better. Therefore, we have now conducted Spearman's rho correlations for the above mentioned associations. 

“Apart from the CFA and Rasch, Spearman’s rho correlations were carried to understand the associations between the MAAS, the ISI, and the GHQ-12 scores.”

“Moreover, the MAAS total score was significantly correlated with the ISI (r=-0.601; p<0.001) and the GHQ-12 (r=-0.384; p<0.001) scores.”

2. - "Western country" instead of "Western countries" (p. 5).

- "Modern test theory" instead of "item response theory" (p. 6).

Reply: We have now edited these. Thanks 

“and becomes outdated in the Western countries,”

“Then, Rasch analysis from the item response theory was applied to evaluate”

---

## [Editor Report · Decision Letter 4]

28 Jun 2021

Using Mindful Attention Awareness Scale on male prisoners: confirmatory factor analysis and Rasch models

PONE-D-20-09274R4

Dear Dr. Pakpour,

We’re pleased to inform you that your manuscript has been judged scientifically suitable for publication and will be formally accepted for publication once it meets all outstanding technical requirements.

Kind regards,

Gilles van Luijtelaar, Ph.D.

Academic Editor

PLOS ONE
---

## [Editor Report · Acceptance letter]

5 Jul 2021

PONE-D-20-09274R4 

Using Mindful Attention Awareness Scale on male prisoners: confirmatory factor analysis and Rasch models 

Dear Dr. Pakpour:

I'm pleased to inform you that your manuscript has been deemed suitable for publication in PLOS ONE. Congratulations! Your manuscript is now with our production department. 

Kind regards, 

on behalf of

Dr. Gilles van Luijtelaar 

Academic Editor

PLOS ONE